# Deletion of a pseudogene within a fragile site triggers the oncogenic expression of the mitotic CCSER1 gene

Benedetta M Santoliquido[1,2], Michela Frenquelli[1], Claudia Contadini[3], Stefano Bestetti[4], Marco Gaviraghi[1], Elisa Barbieri[5], Anna De Antoni[6], Luca Albarello[7], Angelo Amabile[2,8], Alessandro Gardini[5], Angelo Lombardo[2,8], Claudio Doglioni[2,7], Paolo Provero[9,10], Silvia Soddu[3], Davide Cittaro[9], Giovanni Tonon[1,9]

The oncogenic role of common fragile sites (CFS), focal and pervasive gaps in the cancer genome arising from replicative stress, remains controversial. Exploiting the TCGA dataset, we found that in most CFS the genes residing within the associated focal deletions are down-regulated, including proteins involved in tumour immune recognition. In a subset of CFS, however, the residing genes are surprisingly overexpressed. Within the most frequent CFS in this group, FRA4F, which is deleted in up to 18% of cancer cases and harbours the CCSER1 gene, we identified a region which includes an intronic, antisense pseudogene, TMSB4XP8. TMSB4XP8 focal ablation or transcriptional silencing elicits the overexpression of *CCSER1*, through a cis-acting mechanism. CCSER1 overexpression increases proliferation and triggers centrosome amplifications, multinuclearity, and aberrant mitoses. Accordingly, FRA4F is associated in patient samples to mitotic genes deregulation and genomic instability. As a result, cells overexpressing CCSER1 become sensitive to the treatment with aurora kinase inhibitors. Our findings point to a novel tumourigenic mechanism where focal deletions increase the expression of a new class of "dormant" oncogenes.

## Introduction

The cancer genome presents widespread focal deletions (FDs). Whereas a minority of FDs inactivates tumour suppressors, the impact of most FDs remains unknown (Bignell et al, 2010; Chen & Weiss, 2014). A large subset of these FDs occurs at common fragile sites (CFS), chromosomal loci which are prone to breaks elicited by replicative stress (Hazan et al, 2016; Glover et al, 2017). These regions are replicated late during the cell cycle and are enriched in large genes. CFS are pervasive throughout the cancer genome, as they are frequently present in cancers originating from several tissues, although present a remarkable cell type specificity (Beroukhim et al, 2010; Bignell et al, 2010; Zack et al, 2013). Despite their widespread occurrence, their oncogenic role remains controversial (Bignell et al, 2010; Rajaram et al, 2013; Hazan et al, 2016; Glover et al, 2017). Whereas few tumour suppressor genes including *FHIT*, *WWOX* and more recently *PARK2* have been linked to CSFs, a clear role in oncogenesis for the vast majority of genes associated with CFS remains unknown.

## Results

In the attempt to shed light on the role of CFS in oncogenesis, we have exploited the The Cancer Genome Atlas (TCGA) dataset to interrogate the relationship between the FDs occurring at CFS and the expression of the protein coding genes within the CFS boundaries. Out of the 27 CFS included in the analysis (Bignell et al, 2010), several CFS (13 out of 27), despite their genomic spread, included only one gene significantly deregulated (Figs 1A and S1 and Table S1). Most of these genes were down-regulated, in line with their potential role as tumour suppressors. Among these, there were FHIT, WWOX, FANCC (belonging to the Fanconi anaemia pathway), CADM1, and IMMP2L. Interestingly, a recently identified large CFS, FRA6H (Bignell et al, 2010; Fechter et al, 2007), includes the major histocompatibility complex (MHC) region at 6p21.3. We found that the only significantly down-regulated genes within this large region were HLA-A and B genes as well as MHC class I polypeptide–related sequence A (MICA), an NKG2D ligand belonging to the family of stress induced ligands that mediates self-recognition leading to cytotoxicity, and which is frequently down-regulated in cancer through largely unknown mechanisms (Obiedat et al, 2019) (Fig 1B). The deletion of this region may facilitate tumour immune escape through the modulation of the surface expression of MHC class I proteins, as we have recently shown in acute leukaemia (Toffalori et al, 2019).

[1]Functional Genomics of Cancer Unit, Division of Experimental Oncology, Istituti di Ricovero e Cura a Carattere Scientifico (IRCCS) San Raffaele Scientific Institute, Milan, Italy   [2]Vita-Salute San Raffaele University, Milan, Italy   [3]Unit of Cellular Networks and Molecular Therapeutic Targets, IRCCS–Regina Elena National Cancer Institute, Rome, Italy   [4]Protein Transport and Secretion Unit, Division of Genetics and Cell Biology, IRCCS San Raffaele Scientific Institute, Milan, Italy   [5]The Wistar Institute, Gene Expression and Regulation Program, Philadelphia, PA, USA   [6]DNA Metabolism Laboratory, IFOM-The Firc Institute of Molecular Oncology, Milan, Italy   [7]Pathology Unit, IRCCS San Raffaele Scientific Institute, Milan, Italy   [8]San Raffaele Telethon Institute for Gene Therapy (SR-Tiget), IRCCS San Raffaele Scientific Institute, Milan, Italy   [9]Center for Omics Sciences, IRCCS San Raffaele Scientific Institute, Milan, Italy   [10]Department of Neurosciences "Rita Levi Montalcini," University of Torino, Turin, Italy

Correspondence: tonon.giovanni@hsr.it

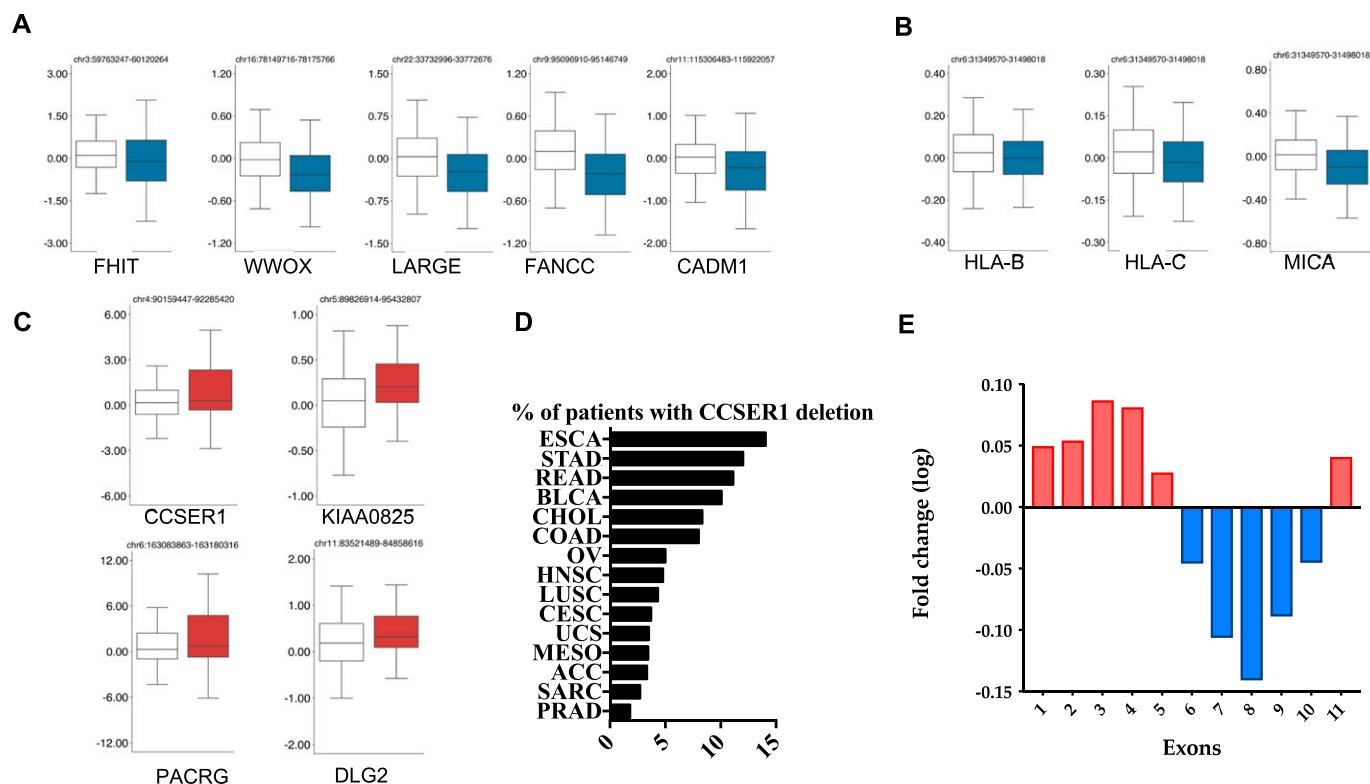

**Figure 1. Deregulated genes within the common fragile site (CFS) in the TCGA dataset.**
**(A)** Box plot of the expression level for genes residing in CFS, comparing for each gene the expression level in tumours without (white) or with (blue) the respective CFS. All the shown genes demonstrate a significant difference in z-scores (Robust Linear Model corrected *P*-value < 1 × 10$^{-2}$). **(B)** Box plots representing the expression level of the genes residing within fragile site FRA6H. **(C)** Box plot for the genes residing in CFS significantly up-regulated. **(D)** Deletion frequency affecting CCSER1 3'portion among tumours (TCGA aCGH data). **(E)** Transcriptomic comparison, calculated for each exon, between patients whose CCSER1 gene presents deletions in the 3'portion of the gene and patients with a CCSER1 wild-type sequence.

Strikingly, FDs in four CFSs led to the unexpected overexpression of the residing genes, namely *DLG2*, *KIAA000825*, *PACRG*, and *CCSER1* (Fig 1C). Whereas the function of KIAA00825 is unknown, DLG2 belongs to the PDZ discs—large polarity protein family which have shown both tumour suppressive as well as oncogenic properties (Roberts et al, 2012). As of *PACRG*, it is remarkable that this gene resides within the FRA6E CFS, where also the tumour suppressor *PARK2* is located, suggesting that the deletion associated with this CFS may concomitantly disrupt *PARK2* and increase the expression of *PACRG*, whose tumourigenic role remains largely unknown.

Finally, *CCSER1* (FAM190A) resides within the CFS FRA4F, on chromosome four at 4q22.1 (Fig S2). After *CDKN2A* and *PTEN*-associated FDs, the genetic lesion occurring at FRA4F represents the most pervasive deletion across tumour types (Glover et al, 2017), affecting especially tumours of the gastroenteric tract, where it involves up to 18% of cases (Fig 1D and Table S2 and [Nancarrow et al, 2008; Bignell et al, 2010; Zack et al, 2013; Glover et al, 2017]).

Given the frequent deletion of FRA4F in cancer, we decided to explore the consequences of this deletion more in detail. CCSER1 features a coil-coiled domain and its down-regulation elicits cell division defects (Patel et al, 2013). The potential tumourigenic role of CCSER1 remains poorly understood. As in other genes involved in CFS (Rajaram et al, 2013; Glover et al, 2017), FRA4F deletions impact only the central portion of CCSER1, thus preserving its 5' and 3'

portions (Scrimieri et al, 2011). We found that tumours presenting CCSER1 deletions presented consistently higher levels of expression in the remaining 5' and 3' portions of the gene, compared with patients with wild-type CCSER1 (Fig 1E).

Seeking to identify a potential mechanism which could explain how a FD in the central portion of *CCSER1* could elicit the increased expression of the remaining part, we explored the genetic elements present within this deleted region. In the intron between exon 8 and 9 resides a pseudogene, *TMSB4XP8*, normally expressed in anti-sense orientation with respect to *CCSER1*. We thus asked whether the selective ablation of *TMSB4XP8* may impact on *CCSER1* expression. To this end, we designed single guide RNAs(sgRNAs) surrounding *TMSB4XP8* and used the CRISPR/Cas9 system to ablate the corresponding genomic region in cells not presenting with *CCSER1* deletion (Figs 2A and S3). TMSB4XP8 deletion led to a significant increase in CCSER1 expression. Conversely, a control sgRNA pair, severing a DNA fragment located nearby within the same intron, had no effect (Fig 2B). We confirmed these data with RNA-FISH, which demonstrated a strong increase in *CCSER1* expression in cells ablated for *TMSB4XP8* (Fig 2C). Taken together, these results suggest that the region where the pseudogene *TMSB4XP8* resides hampers *CCSER1* expression.

We reasoned that two possible mechanisms could underlie this phenomenon. A first hypothesis posits that *TMSB4XP8* RNA may

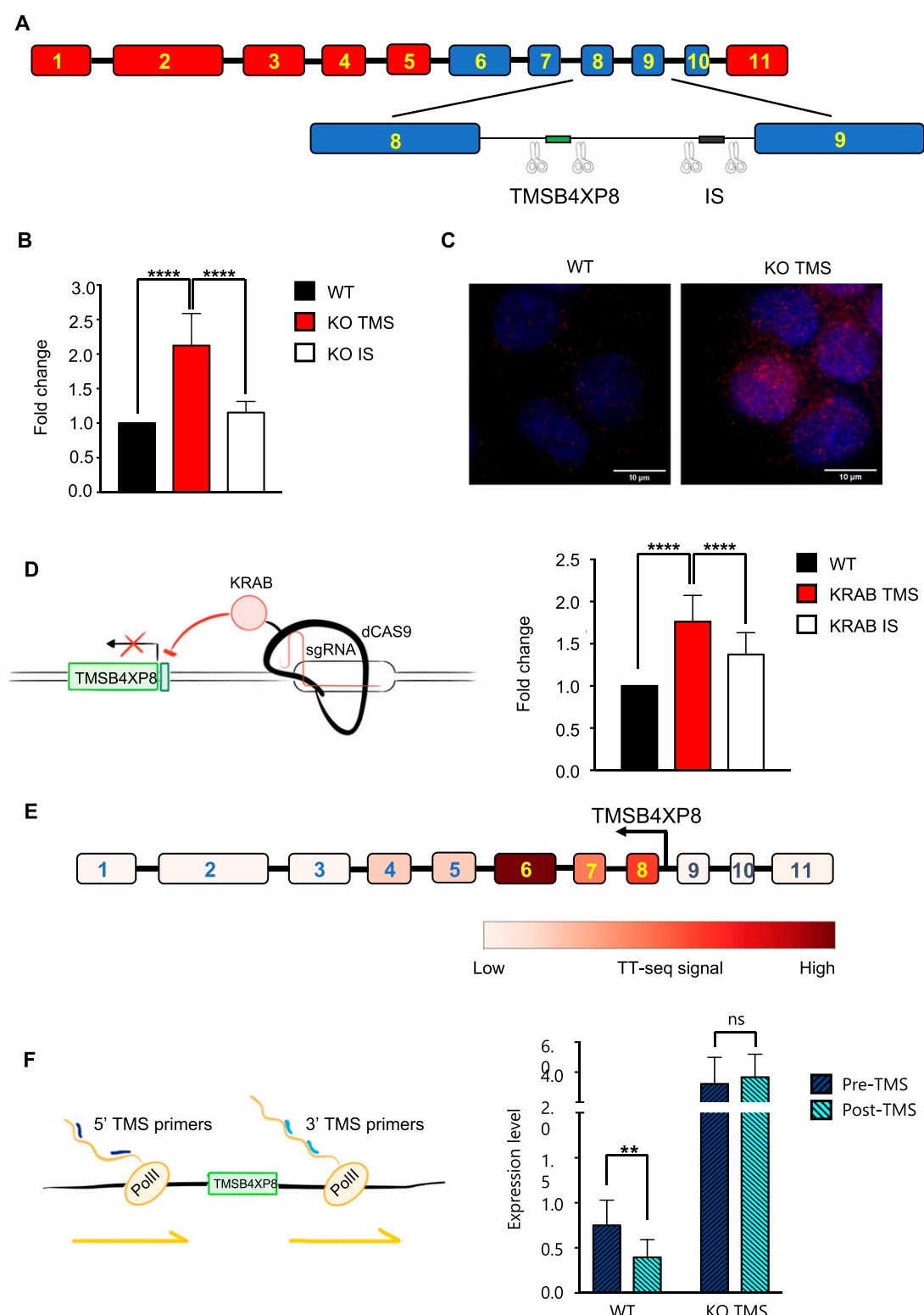

**Figure 2. *TMSB4XP8* ablation increases *CCSER1* expression.**
**(A)** Schematic representation of *CCSER1* gene structure and of the strategy to induce *TMSB4XP8* (TMS) and intronic sequence (IS) KO. **(B)** Expression of CCSER1 in WT, KO TMSB4XP8, and KO IS DLD1 cells analysed by qRT-PCR and expressed as fold change versus WT. The means and SDs of three different experiments are reported. Statistical analysis was performed with two-way ANOVA with Tukey correction. ****P < 0.0001. **(C)** RNA-FISH staining of *CCSER1* mRNA (red) in WT and KO *TMSB4XP8* cells. DAPI was used to stain nuclei. Representative images from two independent experiments. **(D)** Schematic representation of the strategy used to repress *TMSB4XP8* expression and level of expression of *CCSER1* upon *TMSB4XP8* silencing (left). *CCSER1* expression in WT, KRAB TMSB4XP8, and KRAB IS DLD1 evaluated by qRT-PCR and expressed as fold change

directly impede *CCSER1* expression. To test this scenario, we transfected cells with a vector overexpressing the *TMSB4XP8* sequence, yet we could not detect any modification in the expression level of *CCSER1* (data not shown). As an alternative model, we surmised that a cis-acting, local mechanism could underlie the control of *CCSER1* expression by this genomic region. To this end, we exploited an engineered transcriptional repressor strategy that we have recently proposed (Amabile et al, 2016), whereby a repressive Krüppel-associated box (KRAB) domain, linked to the C-terminus of a catalytically inactive Cas9, is targeted to the region of interest through specific sgRNAs. Using this construct, we targeted the promoter region of *TMSB4XP8*. As a result of the silencing of the TMSB4XP8 promoter, the expression of *CCSER1* again increased, suggesting that a cis-acting event occurring at the *TMSB4XP8* promoter might modulate the expression of CCSER1 (Fig 2D).

Gene regulation by antisense transcription is emerging as a central mechanism to control gene transcription in several organisms including bacteria, yeast, and mammalian cellular systems (Pelechano & Steinmetz, 2013). To gain insight whether this mechanism is at play also on the *CCSER1* locus, we exploited published TT-Seq data (Wachutka et al, 2019), a technique which records newly transcribed RNAs. In wild-type cells, where the *TMSB4XP8* sequence is preserved, one would anticipate a drop in the expression of nascent RNA in the exon immediately 3′ from the pseudogene. Indeed, analysis on K562 cell lines, wild type for *CCSER1*, revealed that the expression of *CCSER1* was strongly reduced in the exons 9–11, immediately after the intronic sequence containing TMSB4XP8 (Fig 2E).

We then assessed the levels of *CCSER1* transcript in our cellular models. We hence measured the nascent, chromatin-associated RNA levels of *CCSER1*, 3′ and 5′ with respect to *TMSB4XP8* (Fig 2F). In wild-type cells, the *CCSER1* nascent RNA levels decreased at the 3′ of TMSB4XP8, when compared with the levels at the 5′ side of *CCSER1*. Conversely, the 3′ and 5′ levels of *CCSER1* nascent RNA were higher and also comparable in cells in which *TMSB4XP8* was removed by CRISPR/Cas9.

Taken together, these results suggest that *CCSER1* expression is controlled by *TMSB4XP8* antisense transcription through a local, cis-acting mechanism.

Genomic analyses have predicted that the various genomic rearrangements impacting on the *CCSER1* locus express in-frame *CCSER1* mRNAs (Scrimieri et al, 2011). We hence assessed whether CCSER1 increase in expression is oncogenic. Indeed, overexpression of an isoform isolated from cancer cell lines, lacking a central portion of CCSER1, thus mimicking the situation present in primary tumours (r-CCSER1; Fig S4) was able to increase proliferation in both cancer (HeLa) and immortalized (RPE-hTERT) cell lines (Fig 3A), in line with a recent report (Kang & Park, 2019). To determine the mechanism underlying CCSER1 role in oncogenesis, we relied again on the TCGA dataset, and compared and contrasted the transcriptomic profile of

tumours with or without FDs at *CCSER1*. A functional enrichment analysis revealed that the most frequently altered pathways in tumours where the *CCSER1* locus is deleted were related to mitosis and cell division (Fig 3B). It has been reported that wild-type CCSER1 localizes at the mitotic spindle (Patel et al, 2013). Notably, we found that also the r-CCSER1 isoform, which lacks the only recognized protein domain in CCSER1, the coiled-coil interaction domain, colocalizes to the centrosome, specifically in the pericentriolar material (PCM), during all stages of the cell cycle (Fig 3C).

CCSER1 deficiency has been associated with cell division defects (Patel et al, 2013). Given its localization, and the association of CCSER1 deletion with altered mitotic programs in the TCGA dataset, we next assessed whether overexpression of r-CCSER1 had any impact on the proper execution of mitosis. Indeed, r-CCSER1 overexpression elicited the formation of multinuclear cells (Fig 3D). Time-lapse imaging revealed that cells expressing r-CCSER1 presented pervasive aberrant mitoses (Fig 3E and Video 1), leading to multinuclear cells. As for the mechanism, an assessment of the number of centrosomes per mitosis revealed centrosome amplification upon r-CCSER1 overexpression (Fig 3F).

We next determined the impact of deletions at CCSER1 on chromosomal instability in patient samples. To this end, we used a gene expression signature, which include 25 or 70 genes whose increased expression has been associated with chromosomal instability and functional aneuploidy, in several cancer types (CIN25 and CIN70, respectively) (Carter et al, 2006). This analysis revealed that the genome of patients with CCSER1 deletions was significantly more rearranged than the genome of patients lacking this genetic lesion (Fig 3G) and data not shown.

We reasoned that cells with deletions at the *CCSER1* locus, and increased expression of the rearranged gene, may be more prone to the action of drugs acting on the mitotic machinery. We first confirmed that CRISPR/Cas9-mediated ablation of the pseudogene was associated with increased CCSER1 expression and, in line with the results obtained by CCSER1 overexpression, enhanced proliferation. We then treated CRISPR/Cas9 cells with Danusertib, a broad Aurora kinase inhibitor. The treatment with this compound erased the growth advantage conferred by CCSER1 overexpression (Fig 3H), suggesting that patients with CCSER1 deletions may benefit from the treatment with compounds targeting the mitotic machinery.

## Discussion

In this study, we have identified a novel mechanism of expression regulation that entails the antisense expression of a genetic element which at the chromatin level is able to modulate the expression of a gene involved in mitotic progression. This mechanism is exploited by cancer cells, whereby FDs remove these regulatory elements, thus unleashing the expression of the corresponding

to WT. Mean and SD from three different experiments combined are reported (right). Statistical analysis was performed with two-way ANOVA with Tukey's correction. ****$P$ < 0.0001. Increase in CCSER1 levels in the KRAB IS sample is likely due to KRAB mediates long-range transcriptional repression (Groner et al, 2010). **(E)** Expression levels for each *CCSER1* exons in K562 leukaemia cells (TT-seq data from Wachutka et al [2019]). **(F)** Expression of nascent *CCSER1* RNA evaluated in WT and KO *TMSB4XP8* cells by qRT-PCR. Expression of two different regions (one upstream and one downstream of TMSB4XP8) has been evaluated. Data are presented relative expression ($2^{-\Delta Ct}$) normalized on *GAPDH* expression. Graph shows results from five different experiments combined.

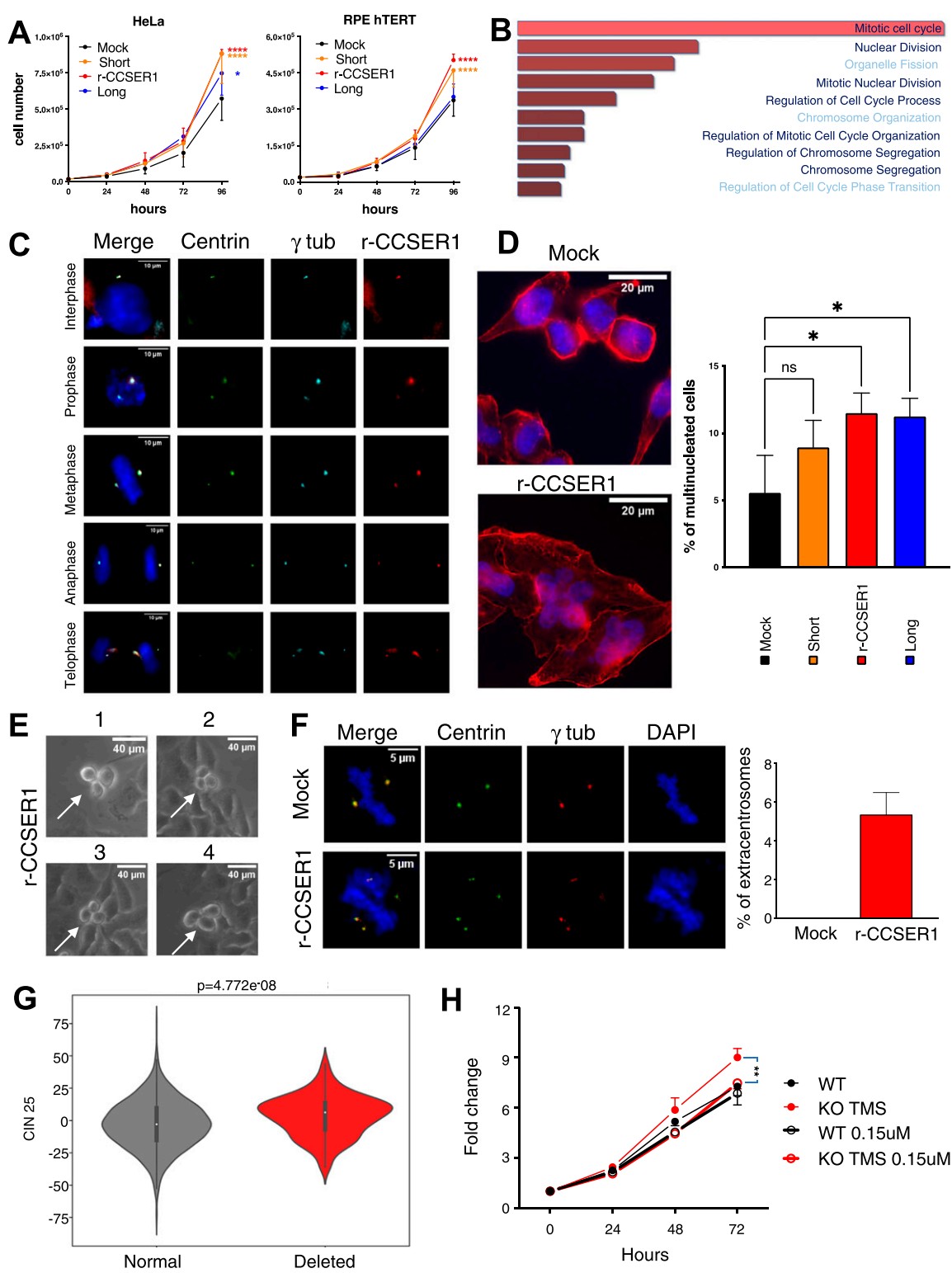

**Figure 3. rCCSER1 overexpression induces an oncogenic phenotype related to mitotic pathway alterations.**
**(A)** Proliferation curves of Hela sgRNA *CCSER1* and RPE-hTERT cells infected with the reported constructs or the mock control. The results shown are the average with SD of three biological replicates. A two-way ANOVA with multiple comparison test was performed for the 96-h dataset. **(B)** Enrichment analysis of pathways deregulated in patients presenting with deleted *CCSER1* versus patients with wild-type *CCSER1*. **(C)** Immunofluorescence staining of Halo-rCCSER1 (red), γ-tubulin (light blue) on HeLa cells stably expressing centrin-GFP evaluated in different phases of cell cycle at 48 h post transfection. DAPI was used to stain cell nuclei. **(D)** Immunofluorescence staining of Phalloidin (red) and of DAPI on HeLa sgRNA cells infected with the reported constructs or the mock control. Staining performed after 2 wk from infection. Two

bona fide oncogene, a phenomenon which seems frequent across cancer types, given the pervasive prevalence of fragile sites in the cancer genome.

We have studied specifically one CFS, FRA4F, which impacts in particular one gene, CCSER1. A previous report has shown how the vast arrays of deletions impacting on CCSER1 oftentimes lead to the formation of in-frame sequences (Scrimieri et al, 2011). Intriguingly, our results suggest that deletions of various regions of the gene (mostly on its 3′) are able to trigger the overexpression of the remaining part of CCSER1, thus driving increased proliferation and mitotic instability. Notably, another report has previously shown how CCSER1 knock-down is associated with mitotic defects (Patel et al, 2013). As for other proteins regulating mitosis, we propose that also the deregulated expression of CCSER1, either in the direction of overexpression or down-regulation, could elicit carcinogenic cell division defects.

There are several examples in various organisms on how genetic elements residing within a gene and oriented in antisense, when expressed, are able to control the expression of the corresponding gene (Pelechano & Steinmetz, 2013; Rosa et al, 2016; Venkatesh et al, 2016). We propose that in cancer, FDs are able to remove genetic elements which in cis keep at bay the expression of "dormant" on-cogenes, such as CCSER1 in the case described herein. It is important to note that of the 6,142 analysed events from TCGA involving CCSER1, 3,155 (51.37%) were also overlapping TMSB4XP8. The deletion of CCSER1, irrespectively on TMSB4XP8 engagement, consistently lead to the increase in CCSER1 expression, suggesting that other mechanisms are involved, either indirectly impacting on TMSB4XP8, or through alternative paths. In fact, within this region, other genetic elements are present, including pseudogenes and ncRNAs, which may impact on CCSER1 expression. In addition, deletions may also affect splicing and ultimately control expression levels (Bryen et al, 2019).

The biological role of pseudogenes has long been questioned. Recent evidence suggest that pseudogenes may exert an important role in modulating the levels of several genes, including tumour suppressor genes, through the competition of miRNAs that would bind to their 3′ end, the so-called ceRNA hypothesis (Salmena et al, 2011). More recently, it has been proposed that cancer pseudogenes may behave as mutagenic factors, impacting on the cellular transcriptional landscape (Cooke et al, 2014; Poliseno et al, 2015). We posit that pseudogenes may exert an additional role, most likely at the transcriptional level, whereby their antisense transcription may reduce the efficiency of polymerase II, thus reducing the mRNA levels of the corresponding gene. It is tempting to speculate that eRNAs, expressed RNA species originating from enhancer sequences (Kim et al, 2010), as well as other regulatory elements endowed with inherent transcriptional activity, may behave similarly to pseudogenes, interfering with the expression of the corresponding antisense gene.

Altogether, our genomic analysis suggests that FDs within CFS may unleash the expression of a new class of "dormant" oncogenes that could become relevant therapeutic targets in a significant fraction of cancer patients.

# Materials and Methods

## Cell culture, transfection, and lentiviral vector

HeLa, RPE-hTERT, and HEK293T cells were grown in DMEM (EuroClone), supplemented with 10% heat-inactivated FBS (EuroClone), and 1% penicillin/streptomycin (EuroClone). DLD1 cells were grown in RPMI-1640 medium, supplemented as above. Transient transfections were performed with FuGENE_HD Reagent (Promega), according to the manufacturer instructions. For lentivirus production, HEK293T cells were transfected with with FuGENE HD Reagent (Promega) as above. To this end, a mix containing 10 $\mu$g of transfer vector, 6.5 $\mu$g of packaging vector ΔR 8.74, 3.5 $\mu$g of Env VSV-G, 2.5 $\mu$g of REV, was added dropwise over a monolayer of HEK293T cells seeded on a 10-cm$^2$ dish. After 16 h, the medium was replaced. 24-h later, the medium containing virus particles was collected and filtered on a 0.45-$\mu$m filter. After infection, RPE hTERT cells selected with 12 $\mu$g/ml puromycin (Life Technologies) for at least 3 d. HeLa cells and DLD1 cells were selected with 1 $\mu$g/ml puromycin (Life Technologies) for at least 3 d. Vectors used for over-expression: pLenti DDK mock, pLenti DDK Rearranged, pHTN HaloTag, p Lenti6 CCSER1 Halo Rearranged, pCDNA3.1-empty, pCDNA3.1-TMSBXP8, LViCas9, pCCLsin.PPT.SV40PA.GFP.mhCMV.SFFV.Krab.Wpre. The LViCas9 transfer plasmid was generated by substituting the Cas9 promoter of the pCW-Cas9 plasmid (No. 50661; Addgene) with the SK-T6 promoter (Amendola et al, 2013).

Cloning procedures and CRISPRs vectors are described below. To assess CCSER1 expression level, it was used Sybr Green PCR Master Mix (Cat. no. 4309155; Thermo Fisher Scientific) and primers: CCSER1 FW GCAACATAAAGCAATAGCGGA and CCSER1 REV: GACTACTGGCAAGAACTTTGGC.

## Constructs cloning

The coding sequence of the rearranged CCSER1 isoform was amplified by rtPCR using RNA isolated from SW620 cell line as a template. The construct includes exons 2 (where the ATG is located) to exons 5 included, and then exon 11 (Fig S4). Notably, the splicing between exon 5 and 11 creates an isoform which is not in frame, thus only a portion, out of frame, of exon 11 is translated. SW620 RNA was initially retrotranscribed using oligo-dT and the SuperScript III

---

representative cells are shown for each staining. Graph shows results from three different experiments combined. At least 600 cells counted for each condition. **(E)** Magnification of representative time-lapse frames of asynchronous HeLa sgRNA CCSER1 infected with rCCSER1. Experiment performed after 1 wk from infection. **(F)** Immunofluorescence staining of γ-tubulin (red) on HeLa cells stably expressing centrin-GFP transfected with rCCSER1 and Mock control evaluated during mitosis after 24 h from transfection. DAPI was used to stain cell nuclei. Two representative cells are shown for each staining. Graph shows results from three different experiments combined. 150 cells counted for each condition. Only mitotic figures showing ectopic/not colocalized centrin/γ tubulin signals were considered. $P$-value < 0.05, $t$ test. **(G)** Box-plot showing chromosomal instability signatures in patients bearing deletion overlapping TMSBX4P8. CIN25 values were extracted from RNA-seq data distributed by TCGA. Samples were stratified according to the deletion overlapping TMSB4XP8 pseudogene. Higher values of CIN25 indicate higher chromosomal instability. $P$-values were calculated using Mann-Whitney U test. **(H)** Fold change curves of CRISPR/Cas9 DLD1 KO TMS and CTR treated or not with 0.15 $\mu$M Danusertib. The results shown are the average with SD of three biological replicates. A two-way ANOVA with multiple comparison test was performed for 72 h dataset.

reverse transcriptase (Invitrogen). Two different fragments of the CCSER1 rearranged cDNA (the first fragment consisting of sequence from ATG to the unique BamHI restriction site contained in CCSER1 sequence, the second fragment by nucleotides between BamHI and the TAA stop codon) were PCR amplified using the Phusion high-fidelity DNA polymerase enzyme (New England Biolabs) with the following primers: CCSER1 ATG-BamHI

FW: CACCATGGGGGACTCAGGAT, CCSER1 ATG-BamHI
REV: ATGGATCCCTTTTACATGTTCCGC, CCSER1 BamH1-TAA
FW: CACCGGATCCATCCTATTTCAGA; CCSER1 BamH1-TAA
REV: TTACCCATCCTGCTGGCCTA.

PCR products were inserted into a pENTR/D-TOPO vector (Invitrogen), digested with BamHI (New England Biolabs) restriction enzyme and ligated to generate the full-length CCSER1 rearranged Open reading frame (ORF). The ligation product was inserted into a pENTR/D-TOPO vector (Invitrogen). Rearranged CCSER1 was then PCR-amplified using the Phusion high fidelity DNA polymerase enzyme (New England Biolabs) with specific primers:

FW: CACCACGCGTCACCATGGGGGACTCAGGATCAAGA and
REV: CTCGAGTCACAATTCGATTTCGATATGTAGCTGT

to subclone it in the pLenti DDK vector using MluI and XhoI cloning sites.

To generate Halo-CCSER1 Rearranged, Halotag sequence was PCR-amplified from pFN21A Long Halo CCSER1 vector (Promega) with specific primers carrying a 5′ NotI consensus tail and devoid of the ORF stop codon (Halo Not I FW: TTGCGGCCGCATGGCAGAAATCGG and Halo Not I REV: AAGCGGCCGCGTTATCGCTCTGAAA) using the Phusion high fidelity DNA polymerase enzyme (New England Biolabs). PCR products were inserted into a pENTR/D-TOPO vector (Invitrogen). The obtained construct and a pDONR221 Gateway vector (Invitrogen) containing CCSER1 Rearranged ORF were digested with NotI restriction enzyme (New England Biolabs) and the fragments of interest were ligated exploiting the T4 DNA ligase enzyme (New England Biolabs) to insert the excised PCR product in frame with CCSER1 ORF cloned downstream a unique NotI site.

CCSER1 rearranged ORF was ultimately transferred to proper expression vectors (pLenti 6) by LR reaction using Gateway LR Clonase II Enzyme Mix (Invitrogen) as described in the manufacturer's instruction. The orientation and the sequences of the inserted fragments were confirmed by Sanger sequencing (Eurofins MWG).

To generate vectors containing sgRNA to erase TMSBXP8 and intronic portion between exon 8 and 9, CRISPRs were selected using the online software CRISPR tool (http://crispr.mit.edu). The sequences used were as follows:

TMSBXP8 FW guide 1 (ACCGTCTCAACTAAGATGTCCCAT);
TMSBXP8 REV guide 1 (AAACATGGGACATCTTAGTTGAGA);
TMSBXP8 FW guide 2 (ACCGTTATGGCGACAACTCGGTGG);
TMSBXP8 REV guide 2 (ACCGTTATGGCGACAACTCGGTGG);
Intron FW guide 1 (ACCGAACCAAGGGGACACTGCCGC);
Intron REV guide1(AAACGCGGCAGTGTCCCCTTGGTT);
Intron FW guide 2 (ACCGTAAACACTATGTAAGGGGGT);
Intron REV guide 2 (AAACACCCCCTTACATAGTGTTTA).

Paired guides were first annealed mixing oligos in equimolar concentrations, heating them at 94°C, and then allowing to cool to room temperature.

The obtained construct and pCCLsin.PPT.pGK.GFP.Wpre.3′ LTR-U6.sgRNA or pCCLsin.PPT.pGK.BFP.Wpre.3′ LTR-U6.sgRNA were digested with BbsI restriction enzyme (New England Biolabs) and the fragments of interest were ligated exploiting the T4 DNA ligase enzyme (New England Biolabs). The sequences of the inserted fragments were confirmed by Sanger sequencing.

## CRISPR/Cas9-mediated genomic deletions and knock-out

To obtain HeLa sgRNA KO for CCSER1, HeLa cells were transfected with plasmid pD1301-AD:154341 expressing both Cas9 and the sgRNA (sgRNA sequence: TGCAAGCCTTTGATTCCCAA **TGG**); a single-cell cloning was performed and each clone was screened for CCSER1 expression.

To obtain DLD1 cells with TMSBXP8 or intronic deletions, DLD1 cells were infected with two different lentivectors; one expressing a sgRNA cutting upstream of TMSBXP8 and the fluorescent marker GFP and the other expressing an sgRNA cutting downstream of TMSBXP8 and the fluorescent marker BFP. The same deletion experiment was performed with the control intronic region.

The sequences of the sgRNA used in those experiments are listed below:

sgRNA TMSBXP8 1 deletion upstream: TCTCAACTAAGATGTCCCAT **GGG**
sgRNA TMSBXP8 2 deletion downstream: TTATGGCGACAACTCGGTGG **TGG**
sgRNA Intronic region 1 deletion upstream: AACCAAGGGGACACTG-CCGC **AGG**
sgRNA Intronic region 2 deletion downstream: TAAACACTATGTAA-GGGGGT **AGG**

(PAM sequence in bold and italic).

Cells positive for both BFP and GFP were selected by cell sorting. LViCas9 was transfected with Fugene HD reagent (Promega) and selected with 1 µg/ml puromycin (Life Technologies) for at least 3 d. PCR was then performed to detect the two deletions (GoTaq Kit; Promega). The primers used to assess TMSBXP8 ablation are:

TMSBXP8 del FW: GCTTTCTGTCATTTCCTGTGC and
TMSBXP8 del REV: GTTCCATGCCCTATCCCATA;

to assess Intronic deletion:

Intronic del FW: CAAAGGCTTTATTCATTCATTCG and
Intronic del REV: TTTTCTGCCCTTAGCTTCCA.

## Proliferation studies and drug experiment

HeLa sgRNA and RPE hTERT, previously infected with pLenti DDK mock and pLenti DDK r-CCSER1 (as described before) were plated in 60-mm dishes in triplicate (20,000 cells/well) with a complete medium. The cells were detached and counted in triplicate at the indicated time points with a Burker cell chamber after Trypan Blue staining to exclude apoptotic/necrotic cells. Where indicated, a two-way ANOVA with a multiple-comparison test was applied. DLD1

cells, previously infected with pCCLsin.PPT.pGK.GFP.Wpre.3′ LTR-U6.sgRNA and pCCLsin.PPT.pGK.BFP.Wpre.3′ LTR-U6.sgRNA and then "crisped" with pLenti Puro CAS9 were plated in a 96-well plate in triplicate (2,000 cells/well) with complete medium. CellTiter-Glo Luminescent Cell Viability Assay is performed at the indicate time points, according to the manufacturer instructions. For drug evaluation, Danusertib (Danusertib [PHA-739358] S1107; Selleck Chemicals) was added to medium 24 h after plating.

### Fluorescence microscopy

Cells were plated on 13-mm coverslips and transfected as reported. After 48 h, cells were washed twice with PBS, fixed for 10 min with 2% formaldehyde, and washed in PBS. Permeabilization was performed with 0.25 Triton X-100 (Sigma-Aldrich) in PBS for 5 min and washed in PBS. Then cells were fixed for a second time with ice-cold 100% methanol for 10 min and washed twice in PBS. For immunofluorescence staining, coverslips were washed twice with PBS-TWEEN 0.05% (Sigma-Aldrich) and blocked with 3% BSA supplied with 0.1% Triton (Sigma-Aldrich) in PBS (30 min in a wet and dark chamber). Coverslips were washed twice with PBS-Tween 0.05% and incubated with primary antibodies for 2 h at 37°C in a wet and dark chamber. Coverslips were washed in PBS-Tween 0.05% and water and incubated with secondary antibodies for 1 h at 37°C in a wet and dark chamber. Coverslips were washed with PBS-Tween 0.05% and water and mounted on microscope slides using ProLong Gold Antifade Reagent with DAPI (Invitrogen). The antibodies used were: γ-Tubulin (T5326, 1:1,000; Sigma-Aldrich); Alexa Fluor 546–conjugated Phalloidin (Cat. N. A22283; Molecular Probes). Moreover, we used the HaloTag TMRDirect Ligand (G299A 1:10,000; Promega) added to medium before fixation. Coverslips were then stained with Alexa Fluor 647–conjugated α-mouse (1:800; Invitrogen). HeLa GFP centrin-2 cells were analysed for localization and aberrant number of centrosomes with Leica SR GSD 3D total internal reflection fluorescence (160×), for multinuclearity and for counting number of cells with aberrant centrosomes with AxioImager A2 fluorescent microscope (Zeiss) (100×). Quantifications of plurinucleated cells were performed by counting at least 200 cells per condition in each experiment, quantifications on centrosomes amplification were performed by counting at least 50 cells.

### Time-lapse microscopy

Cells were seeded in six-well plates coated with 0.02 mg of Poly-L-ornithine solution (Sigma-Aldrich). Time-lapse experiments were performed taking advantage of the 20× wide-field imaging of a Zeiss Axiovert S100 TV2 equipped with a thermostated chamber to maintain a temperature of 37°C and 5% CO2 concentration for the entire duration of the experiment. Every 5 min an image of the same field was acquired. The acquisition was performed for a total of 24 h.

### TMSB4XP8 overexpression

To overexpress the TMSB4XP8 pseudogene, DLD1 cells, either CTR or TMSB4XP8 KO, were transfected with a pCDNA3.1 vector containing the FLAG-tagged TMSB4XP8 sequence (Genscript) or with the

pCDNA3.1 empty vector as control. Transfection was performed with Fugene-HD (Promega) following the manufacturer's protocol. 48 h after transfection, RNA was extracted and cDNA synthetized with the ImProm II-Reverse transcription System (Promega). Expression of CCSER1 and TMSB4XP8-FLAG was assessed by Real-Time PCR using Sybr Green master Mix (Thermo Fisher Scientific) and the following primers:

CCSER1 FW: GCAACATAAAGCAATAGCGGA
CCSER1 REV: GACTACTGGCAAGAACTTTGGC
TMSB4XP8 FW: GACAAACCCGATATGGCTGA
TMSB4XP8-FLAG REV: CTTGTCGTCATCGTCTTTGTAGTC

### Subcellular fractionation and chromatin-bound RNA extraction

Subcellular fractionation was performed as follows. Cells were washed with ice-cold PBS, resuspended in cold lysis buffer with 0.15% NP-40, and the lysate layered on sucrose buffer to isolate nuclei. Glycerol buffer (20 mM Tris, pH 7.9, 75 mM NaCl, 0.5 mM EDTA, 50% glycerol, and 0.85 mM DTT) and nuclei lysis buffer (20 mM Hepes, pH 7.6, 7.5 mM MgCl$_2$, 20 mM EDTA, 0.3 M NaCl, 1 M urea, 1% NP-40, and 1 mM DTT) were used to isolate nucleoplasmic fraction and chromatin-bound RNA fraction. Chromatin-bound RNA was isolated with Trizol protocol (Life Technologies) and further purified with RNAesy mini-kit (QIAGEN) after DNAse digestion (QIAGEN). Ribosomal RNA was removed with RiboMinus Eukaryote Kit (Life Technologies).

Expression of immature CCSER RNA was evaluated by real-time PCR with the following primers:

Pre-TMS FW: CAAGTGTTCATCCCCTAACTTTGA
Pre-TMS REV: GAAAAACAGCGGCCAAGTG
Post-TMS FW: GCTACAGCTTCATTGTTGCATC
Post-TMS REV: CAGGGCTTAGGACCTGCTT
GAPDH Intron1 FW: AGACGGGCGGAGAGAAAC
GAPDH Intron1 REV: CGGAGGGAGAGAACAGTGAG

### RNA-FISH

Custom Stellaris FISH Probes were designed based on CCSER1 sequence by using the Stellaris RNA FISH Probe Designer (Biosearch Technologies, Inc.; www.biosearchtech.com). The cells were hybridized with CCSER1 Stellaris RNA FISH Probe set labeled with Q570 dye (Biosearch Technologies, Inc.), following the manufacturer's instructions (www.biosearchtech.com/stellarisprotocols). Briefly, cells were plated on 13-mm coverslips, and 24 h later, cells were washed twice with PBS, fixed for 10 min with 4% formaldehyde, and washed in PBS. Permeabilization was performed with 0.1% Triton X-100 (Sigma-Aldrich) in PBS for 10 min. Cells were then incubated for 5 min in Buffer A (2× SSC, formamide 10%) before the incubation with CCSER1-specific probes (overnight in a wet and dark chamber). Coverslips were washed twice with Buffer A for 30 min at 37°C, twice with PBS, and then nuclei were stained with DAPI for 5 min, washed twice with PBS, and once with Buffer B (2× SSC). Coverslips were finally mounted on microscope slides using Mowiol. Images were acquired with DeltaVision Ultra microscope (GE HealthCare).

### Bioinformatic analysis

Segmentation data were downloaded from TCGA (grch38.seg.v2 files, derived from Affymetrix SNP 6.0 platform). Data were filtered for a segmentation value lower than –0.1. Patient-wise segmentation value over the CFSs was then calculated using bedtools (Quinlan, 2014). RNA-seq data expressed as FPKM were downloaded from the TCGA portal. For each tumour type, we calculated genewise Z-scores estimating mean and standard deviation from the expression values in normal tissue. We investigated the relationship between the presence of a deletion and the underlying gene expression by Robust Linear Model using Tukey's biweight function to trim outliers. We considered the estimate of tumour purity (Aran et al, 2015) as an additional covariate in our linear model. *P*-values were corrected using Bonferroni's procedure. Analysis of CFS has been then restricted to relationships involving annotated protein coding genes with an adjusted *P*-value lower than 0.01.

### TT-seq analysis

We downloaded TT-seq data in bigwig format from the original authors' page (https://i12g-gagneurweb.informatik.tu-muenchen.de/public/TTSeqBigWig/) (Wachutka et al, 2019) data represent two replicates in seven time points. We extracted the signal over CCSER1 exons divided by exon lengths ($E_i$ for $i$ = 14 samples), we calculated exon expression as $\log(1 + \sum E_i)$ and rescaled it in the range [0, 1].

# Supplementary Information

# Acknowledgements

We are grateful to D Gabellini, V Pelechano, R Bernardi, M Muzio, A Cantore, and members of the Tonon lab for sharing reagents and critical reading of the manuscript. We also thank Livia Caizzi for her precious support with the TT-seq dataset. Part of this work was carried out in ALEMBIC, an advanced microscopy laboratory established by IRCCS Ospedale San Raffaele and Università Vita-Salute San Raffaele. This work was supported by the Associazione Italiana per la Ricerca sul Cancro (AIRC; Investigator Grants #21360) (G Tonon). This work was supported by Fondazione AIRC (Investigator Grant #21360) (G Tonon).

## Author Contributions

BM Santoliquido: conceptualization, investigation, methodology, and writing—original draft, review, and editing.
M Frenquelli: conceptualization, investigation, methodology, and writing—review and editing.
C Contadini: formal analysis, investigation, and methodology.
S Bestetti: software, formal analysis, validation, and visualization.
M Gaviraghi: investigation and methodology.
E Barbieri: data curation, formal analysis, and methodology.
A De Antoni: data curation, formal analysis, and methodology.
L Albarello: data curation and formal analysis.
A Amabile: data curation and formal analysis.
A Gardini: conceptualization and methodology.
A Lombardo: conceptualization, data curation, and formal analysis.
C Doglioni: conceptualization, data curation, formal analysis, investigation, and methodology.
P Provero: conceptualization, data curation, formal analysis, and validation.
S Soddu: conceptualization, formal analysis, and investigation.
D Cittaro: data curation, formal analysis, investigation, and methodology.
G Tonon: conceptualization, resources, data curation, formal analysis, supervision, funding acquisition, investigation, visualization, methodology, project administration, and writing—original draft, review, and editing.

## Conflict of Interest Statement

The authors declare that they have no conflict of interest.

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
