## [Reviewer comments · Life Science Alliance]

Life Science Alliance

Deletion of a pseudogene in a fragile site triggers oncogenic expression of the CCSE1 mitotic gene

Benedetta Santoliquido, Michela Frenquelli, Claudia Contadini, Stefano Bestetti, Marco Gaviraghi, Elisa Barbieri, Anna De Antoni, Luca Albarello, Angelo Amabile, Alessandro Gardini, Angelo Lombardo, Claudio Doglioni, Paolo Provero, Silvia Soddu, Davide Cittaro, and Giovanni Tonon
DOI: <https://doi.org/10.26508/lsa.202101019>

Corresponding author(s): Giovanni Tonon, San Raffaele Scientific Institute

Review Timeline:

Submission Date:	2021-01-13
Editorial Decision:	2021-03-24
Revision Received:	2021-04-20
Editorial Decision:	2021-05-24
Revision Received:	2021-06-12
Accepted:	2021-06-14

Transaction Report:

March 24, 2021

Re: Life Science Alliance manuscript #LSA-2021-01019-T

Dr. Giovanni Tonon
San Raffaele Scientific Institute
Functional Genomics of Cancer Unit, Division of Molecular Oncology
Via Olgettina 60
Milan 20032
Italy

Dear Dr. Tonon,

Thank you for submitting your manuscript entitled "Deletion of a pseudogene within a fragile site triggers the oncogenic expression of the mitotic CCSE1 gene" to Life Science Alliance (LSA). The manuscript was assessed by expert reviewers, whose comments are appended to this letter.

We apologize for this extended and unusual delay in getting back to you. As you will note from the reviewers' comments below, the reviewers are intrigued by these findings, but have also raised few reasonable concerns that should be addressed prior to further consideration of this manuscript at LSA. We would, thus, encourage you to submit a revised version of this study to LSA that addresses all of the reviewers' points.

Thank you for this interesting contribution to Life Science Alliance. We are looking forward to receiving your revised manuscript.

Sincerely,

Shachi Bhatt, Ph.D.

Executive Editor

Life Science Alliance

<https://www.lsjournal.org/>

Interested in an editorial career? EMBO Solutions is hiring a Scientific Editor to join the international Life Science Alliance team. Find out more here -

https://www.embo.org/documents/jobs/Vacancy_Notice_Scientific_editor_LSA.pdf

B. MANUSCRIPT ORGANIZATION AND FORMATTING:

Reviewer #1 (Comments to the Authors (Required)):

The manuscript by Santoliquido and colleagues reports the effect of deletions within a common fragile site (FRA4F) on the expression of CCSE1, a large gene residing at this locus, and its potential oncogenic role linked to its function in mitosis. They show that deletion or transcriptional

repression of a pseudogene located in an antisense orientation in the intron between exons 8 and 9 of CCSER1, results in the increased expression of CCSER1 and leads to mitotic defects, notably supernumerary centrosomes and multinucleation.

While the premise is interesting, there are several points that, in my opinion, should be addressed by the authors to better support their conclusions and the significance of their study.

The authors first analyse transcriptomic data from TCGA to find a correlation between the expression of CFS genes with respect to focal deletions at CFSs in cancer. They refer to 27 CFS and to the work of Bignell GR et al., Nature 2010. However, in their work Bignell et al. found 19 homozygous deletion clusters overlapping the 39 finely mapped fragile sites. In addition, some of the "unexplained" HD clusters are in large genes (MACROD2 and NEGR1) that have been characterized as CFSs by Le Tallec et al., Cell Reports, 2013).

- Could the authors better explain and describe the CFS included in the analysis?
- The coordinates for each CFS and the corresponding genes should also be reported in Figure 1.
- Some of the genome coordinates seem not to correspond: for instance, FANCC genomic location is chr9:95,099,054-95,426,796 (GRCh38/hg38) chr9:97,861,336-98,079,991 (GRCh37/hg19) while the HD cluster position mapped at FRA9D is 96,899,013-96,948,852 (Supplementary Table 7, Bignell et al., 2010). Could the authors explain this apparent discrepancy?
- The authors state in the text that FRA4F deletions impact only the central portion of CCSER1, thus preserving its 5' and 3' portions (Scrimieri et al., 2011). However in the legend of Figure 1d,e they analyse the deletion frequency affecting the 3' portion. Then, they analyse the effects of deletion of TMSBX4P8 pseudogene that resides between exons 8 and 9. In the study of Scrimieri et al., it is shown that FAM190A (CCSER1) rearrangements are characterized by in-frame deletions involving several different exons. This leads to the expression of various mutant transcripts, which may provide new, or possibly convey dominant-negative functions. Which is the proportion of deletions overlapping TMSBX4P8? In case of rearrangements that do not involve TMSBX4P8, what is the mechanism leading to CCSER1 over-expression? Do the CCSER1 over-expression result in the same effect of expression of mutant isoforms?
- The authors compare the phenotype elicited by CCSER1 over-expression with over-expression of an isoform isolated from cancer cell lines (r-CCSER1). Could the authors describe the structure of this isoform? Why the authors do not compare in the same cell line the phenotype associated with TMS ablation or r-CCSER1 over-expression?
- Similarly, since genomic lesions in primary tumors may result in transcripts with different in-frame deletions, does the treatment with the Aurora kinase inhibitor produce the same effect?
- How the authors selected the dose of Aurora kinase inhibitor? Did they perform a dose-response assay?

Minor concerns:

- The authors should clarify whether the results obtained in cells bearing the TMS deletion correspond to a single clone or a cell population.

- Figure 2f: it would be useful to specify the position of the primers with respect to the exon/intron. Also, the figure legend should describe how the values of the RNA level in the graph were calculated.
- Figure 3d: is the graph reporting the % of multinucleated cells? If so, this should be indicated in the y axis. Is the difference between mock and r-CCSER significant?
- Figure 3f: is the graph reporting the % of cells with extracentrosomes? It looks strange to this reviewer that the basal level of cells with extracentrosomes in mock-treated HeLa cells is 0%.
- Suppl. Fig. 2: what is the expected size of the amplicon?

Reviewer #2 (Comments to the Authors (Required)):

The study by Santoliquido examined the TCGA dataset to interrogate the relationship between the focal deletions occurring at Common Fragile Sites (CFS) and the expression of the protein-coding genes for 27 CFS. Among 27 CFS, 13 CFS harbor giant genes that span the entire CFS. Unexpectedly, four CFS showed the association between the focal deletion and the increased expression of the gene within CFSs. CCSER1 at FRA4H exhibited focal deletions across tumor types. Authors investigated the mechanisms by employing the CRISPR/CAS9 mediated ablation of the pseudogene TMSB4XP8 that is transcribed in the antisense orientation. The ablation led to the overexpression of CCSER1 and hyperproliferation. The authors also proposed that TMSB4XP8 regulates the mRNA of CCSER1 by an antisense-mediated mechanism. Rearranged, tumor-form CCSER1 localized to the centrosome and mitotic aberrations seemed prominent in CCSER1-overexpressed cells.

Major critiques

CCSER1 at FRA4H has been studied in the past and the association between focal deletions and variants that skip some of the exons, including the deleted exons, was shown. The authors need to introduce those studies in the introduction, and discuss whether this study supported the results from previous studies or reached different conclusions.

Authors employed CRISPR/CAS9 mediated intervention of TMSB4XP8 in Figure 2b-d, either by deleting or transcriptionally repressing the TMSB4XP8. If these experiments were successful, one would anticipate the genomic deletion of TMSB4XP8 or reduced TMSB4XP8 mRNA. Genomic deletions were presented in the supplemental figure 2 to show that deletions occurred at least in some fraction of cells. However, it was a qualitative, not quantitative analysis. How efficiently the TMSB4XP8 and control regions were deleted remained unclear, which is crucial for assessing the role of TMSB4XP8 in the regulation of CCSER1 expression. qPCR experiments with primers targeting either the deleted allele or wild type allele should tell the fraction of cells that undergo deletion. For the KRAB mediated intervention, you should be able to assess the expression of TMSB4XP8 by qPCR.

It is unclear whether the observed pattern of CCSER1 nascent RNA (that the exons in the upstream of TMSB4XP8 were more abundant than the downstream exons) fits with the antisense-mediated cis-acting mechanism. Do you propose that the binding of antisense RNA to nascent RNA prevents the elongation of mRNA past TMSB4XP8? Then, why were the expression levels of

exons 1-3 lower than those of 6-8? Please provide details in your model and cite (and discuss) the appropriate references for the proposed mechanism.

In Fig. 3, the authors introduced the rearranged forms of CCSER1 (rCCSER1) into cells and tested the cell proliferation and other phenotype conferred by rCCSER1. Please provide the exact structure of rCCSER1. Which are the missing exons? The authors presented the data of Aurora Kinase inhibitor in Fig. 3h. Unlike Fig. 3a, the experiments were terminated at 72 hours. Since the difference was small and KO TMS cells with the inhibitor seemed to accelerate the growth after 48 hours, this reviewer requests the experiment with an extended period of time, up to 96 hours, the same as Fig. 3a.

Other comments

Abstract

"FRA4F, which is present in up to 18% of cancer cases" - either "which is expressed" or "which is deleted." FRA4F is present in the human genome.

Results and Discussion

Page 3, "only one gene significantly deregulated" - "significantly deregulated" here seems redundant, as you mentioned it in the next sentence.

Figure 1 - in somewhere, please provide the genomic view (UCSC genome browser view) of FRA4F, so that we can understand how large the CFS (and CCSER1 gene) is.

Figure 1d - please spell out all the tumor types somewhere in the legend.

Page 5, "this genetic lesion represents the most pervasive" - "lesion" should be "region"

Page 5, "Given the widespread involvement of FRA4F in cancer" - unclear. "Given the frequent deletion of FRA4F in cancer"

Page 6, "To this end, we designed sgRNA probes" - "probes" is not necessary.

Page 9, "preserve the expression of in-frame CCSER1 mRNA" - unclear. "express the CCSER1 in-frame variants"

Figure 3a, can you make the Fig. 3a bigger? The sizes of the font in 3a are too small, compared to other figures in Fig. 3.

Figure 3c, gammatub needs attention.

Page 12, "CCSER1 deficiency has been associated with cell division defects" - Cell cycle profiling by FACS would be powerful to strengthen your argument.

Page 12, Please explain briefly what are the CIN25 and CIN70 better. Also, only CIN25 is presented in Fig. 3G.

Milan, April 19, 2021

Dear Dr. Bhatt,
many thanks for your assistance, and for the Reviewers for their constructive criticisms.
Please find below the responses to their observations, alongside new data.

Many thanks again,
Kind regards,
Giovanni Tonon

Reviewer #1:

The manuscript by Santoliquido and colleagues reports the effect of deletions within a common fragile site (FRA4F) on the expression of CCSER1, a large gene residing at this locus, and its potential oncogenic role linked to its function in mitosis. They show that deletion or transcriptional repression of a pseudogene located in an antisense orientation in the intron between exons 8 and 9 of CCSER1, results in the increased expression of CCSER1 and leads to mitotic defects, notably supernumerary centrosomes and multinucleation.

While the premise is interesting, there are several points that, in my opinion, should be addressed by the authors to better support their conclusions and the significance of their study.

The authors first analyse transcriptomic data from TCGA to find a correlation between the expression of CFS genes with respect to focal deletions at CFSs in cancer. They refer to 27 CFS and to the work of Bignell GR et al., Nature 2010. However, in their work Bignell et al. found 19 homozygous deletion clusters overlapping the 39 finely mapped fragile sites. In addition, some of the "unexplained" HD clusters are in large genes (MACROD2 and NEGR1) that have been characterized as CFSs by Le Tallec et al., Cell Reports, 2013).

- Could the authors better explain and describe the CFS included in the analysis?

We thank the Reviewer for this important observation. We now have specified better in the Material and Methods that analysis of CFS has been then restricted to relationships involving annotated protein coding genes with an adjusted p-value lower than 0.01. Information on all CFS are included in Supplementary Table 1, including on the CFSs not included in the final analysis (we indicated in red all the added/modified text, in the main manuscript).

- The coordinates for each CFS and the corresponding genes should also be reported in Figure 1.

We have replaced the panels in figure 1, including now also the coordinates.

- Some of the genome coordinates seem not to correspond: for instance, FANCC genomic location is chr9:95,099,054-95,426,796 (GRCh38/hg38)

chr9:97,861,336-98,079,991 (GRCh37/hg19)

while the HD cluster position mapped at FRA9D is 96,899,013-96,948,852 (Supplementary Table 7, Bignell et al., 2010). Could the authors explain this apparent discrepancy?

The coordinates in the original table were reported on hg18 reference genome. In order to work with TCGA data we performed a liftOver operation to the hg38 build.

- The authors state in the text that *FRA4F* deletions impact only the central portion of *CCSER1*, thus preserving its 5' and 3' portions (Scrimieri et al., 2011). However in the legend of Figure 1d,e they analyse the deletion frequency affecting the 3' portion. Then, they analyse the effects of deletion of *TMSBX4P8* pseudogene that resides between exons 8 and 9. In the study of Scrimieri et al., it is shown that *FAM190A* (*CCSER1*) rearrangements are characterized by in-frame deletions involving several different exons. This leads to the expression of various mutant transcripts, which may provide new, or possibly convey dominant-negative functions. Which is the proportion of deletions overlapping *TMSBX4P8*? Out of the 6142 analyzed events from TCGA involving *CCSER1*, 3155 (51.37%) are also overlapping *TMSB4XP8*. It should be mentioned that we found that deletion of *CCSER1*, irrespectively on *TMSB4XP8* engagement, consistently lead to the increase in *CCSER1* expression, suggesting that other mechanisms are involved, either indirectly impacting on *TMSB4XP8*, or through alternative paths.

In case of rearrangements that do not involve TMSBX4P8, what is the mechanism leading to CCSER1 over-expression?

This is an interesting point, however at this point we have not assessed the potential role of other deletions on this regard. As mentioned above, deletions in this region consistently lead to increased *CCSER1* expression. Within the boundaries of the deletions not encompassing *TMSB4XP8* we noticed other pseudogenes and ncRNAs, some in antisense orientation, whose deletion may increase *CCSER1* expression though a mechanism similar to the one described for *TMSB4XP8*.

Do the CCSER1 over-expression result in the same effect of expression of mutant isoforms?

We thank the Reviewer for raising this important point. In cells not presenting deletions at this locus, *CCSER1* exists in two isoforms, a short and a long isoform. We attempted to overexpress also these wild type *CCSER1* isoforms, alongside the mutant isoform, obtaining similar results (see figure below, one representative experiment of 3). We posit that the deregulated expression of *CCSER1*, by itself, is able to increase proliferation, irrespectively from the respective isoform engaged.

[Figure removed by editorial staff per authors' request]

- The authors compare the phenotype elicited by *CCSER1* over-expression with over-expression of an isoform isolated from cancer cell lines (*r-CCSER1*). Could the authors describe the structure of this isoform?

We isolated the isoform from the SW-620 cell line (below an IGV representation obtained from the CCLE dataset). We would be happy to include also the corresponding sequence as Supplementary data.

[Figure removed by editorial staff per authors' request]

Why the authors do not compare in the same cell line the phenotype associated with TMS ablation or r-CCSER1 over-expression?

The Reviewer raises a very good point, which we should have explained better in the text. We choose HeLa cells because this is the cell line used in the mitotic field. We found that HeLa cells presented very low levels of CCSER1. To obtain a model completely knocked out for CCSER1, we removed by CRISPR/Cas9 all expressed CCSER1 from the cells (figure below, from a Blast search, where the gRNA sequence is indicated as Query_54741, and the beginning of the CCSER1 gene is indicated in red, ATG sequence):

Conversely, for the drug treatment with aurora kinase inhibitors, we choose a cell line where CCSER1 was already present, and assessed whether a further increase on the gene expression, as triggered by the ablation of TMSB4XP8 via CRISPR/Cas9, could lead to increased proliferation and sensitivity to aurora kinase inhibitors.

- *Similarly, since genomic lesions in primary tumors may result in transcripts with different in-frame deletions, does the treatment with the Aurora kinase inhibitor produce the same effect?*

We thank the Reviewer for this interesting observation. Based on the data presented above, where each CCSER1 isoform was able to increase proliferation, we would argue that the overexpression of CCSER1 transcripts, irrespectively from the size of the in-frame deletion, would always lead to increased proliferation and sensitivity to Aurora kinase inhibitors, as even the WT isoforms trigger mitotic instability upon overexpression (we would be happy to include these data as well).

[Figure removed by editorial staff per authors' request]

- *How the authors selected the dose of Aurora kinase inhibitor? Did they perform a dose-response assay?*

We did perform a dose-response curve (ranging from 0,10 uM to 4,5 uM) and selected the minimal dose that was effective on TMS-KO cells and not on WT cells, with the aim to assay the compound at a dosage which was not toxic to normal cells. It should be also noted that an inhibitor for aurora kinase A, Alisertib, was not able to control the increased proliferation elicited by CCSER1 overexpression, suggesting that only a broad aurora kinase inhibitor may impact on cells overexpressing CCSER1.

Minor concerns:

- *The authors should clarify whether the results obtained in cells bearing the TMS deletion correspond to a single clone or a cell population.*

We thank the Reviewer for this important observation. We confirm that the results obtained are derived from a cell population.

- *Figure 2f: it would be useful to specify the position of the primers with respect to the exon/intron. Also, the figure legend should describe how the values of the RNA level in the graph were calculated.*

We apologize for the mislabelling, the primers were reported in the Method section with the name "Intron1" and "Intron2". We have now renamed the primers in accordance with what reported in figure 2f. The genomic coordinates of the primers used, which we report below, are:

Pre-TMS primers:

90836776 90836991

Post-TMS primers:

90839487 90839581

RNA levels are expressed as relative expression ($2^{-\Delta Ct}$) normalized on GAPDH expression. The figure legend has been modified accordingly.

- *Figure 3d: is the graph reporting the % of multinucleated cells? If so, this should be indicated in the y axis. Is the difference between mock and r-CCSER significant?*

The Reviewer is right, the graph should show as below. Also, the difference in the % of multinucleated cells is indeed significant ($p < 0.05$).

We have now corrected the figure and indicated the significance.

[Figure removed by editorial staff per authors' request]

- *Figure 3f: is the graph reporting the % of cells with extracentrosomes? It looks strange to this reviewer that the basal level of cells with extracentrosomes in mock-treated HeLa cells is 0%.*

We thank the Reviewer for pointing this out. In our experiments we noticed an increase in the centrin signal in the r-CCSER1 overexpressing cells, which was not present in cells transfected with the mock, which instead presented only properly organized centrosomes, where the centrin and gamma-tubulin always colocalized. Notwithstanding, we agree with the Reviewer and would be happy to remove the graph, mentioning the percentage in the main text.

- *Suppl. Fig. 2: what is the expected size of the amplicon?*

We apologize for not having specified before.

The expected amplicon sizes are 627 bp and 422 bp for pseudogene region and intronic sequence region, respectively. We have modified the figure, and the figure legend, accordingly.

Reviewer #2:

The study by Santoliquido examined the TCGA dataset to interrogate the relationship between the focal deletions occurring at Common Fragile Sites (CFS) and the expression of the protein-coding genes for 27 CFS. Among 27 CFS, 13 CFS harbor giant genes that span the entire CFS. Unexpectedly, four CFS showed the association between the focal deletion and the increased expression of the gene within CFSs. CCSER1 at FRA4H exhibited focal deletions across tumor types. Authors investigated the mechanisms by employing the CRISPR/CAS9 mediated ablation of the pseudogene TMSB4XP8 that is transcribed in the

antisense orientation. The ablation led to the overexpression of CCSER1 and hyperproliferation. The authors also proposed that TMSB4XP8 regulates the mRNA of CCSER1 by an antisense-mediated mechanism. Rearranged, tumor-form CCSER1 localized to the centrosome and mitotic aberrations seemed prominent in CCSER1-overexpressed cells.

Major critiques

CCSER1 at FRA4H has been studied in the past and the association between focal deletions and variants that skip some of the exons, including the deleted exons, was shown. The authors need to introduce those studies in the introduction, and discuss whether this study supported the results from previous studies or reached different conclusions.

We thank the Reviewer for her/his observation. As the study on this FCS emerged after the screening of the TCGA data, we propose to add this information in the result and discussion section (we indicated in red all the added/modified text, in the main manuscript).

Authors employed CRISPR/CAS9 mediated intervention of TMSB4XP8 in Figure 2b-d, either by deleting or transcriptionally repressing the TMSB4XP8. If these experiments were successful, one would anticipate the genomic deletion of TMSB4XP8 or reduced TMSB4XP8 mRNA. Genomic deletions were presented in the supplemental figure 2 to show that deletions occurred at least in some fraction of cells. However, it was a qualitative, not quantitative analysis. How efficiently the TMSB4XP8 and control regions were deleted remained unclear, which is crucial for assessing the role of TMSB4XP8 in the regulation of CCSER1 expression. qPCR experiments with primers targeting either the deleted allele or wild type allele should tell the fraction of cells that undergo deletion. For the KRAB mediated intervention, you should be able to assess the expression of TMSB4XP8 by qPCR.

We have assessed CRISPR/Cas9 efficiency just by qualitative PCR, not by qPCR.

As for the expression of TMSB4XP8, there are several homologs of TMSB4XP8 throughout the genome, which differ marginally among themselves (3 nucleotides at most), so despite several efforts we have not been able to separate TMSB4XP8 from the others.

It is unclear whether the observed pattern of CCSER1 nascent RNA (that the exons in the upstream of TMSB4XP8 were more abundant than the downstream exons) fits with the antisense-mediated cis-acting mechanism. Do you propose that the binding of antisense RNA to nascent RNA prevents the elongation of mRNA past TMSB4XP8? Then, why were the expression levels of exons 1-3 lower than those of 6-8? Please provide details in your model and cite (and discuss) the appropriate references for the proposed mechanism.

We wish to thank the Reviewer for giving us the opportunity to explain more properly our proposed model. We do not think that the antisense RNA to nascent RNA may interfere with the elongation of CCSER1 mRNA, since the overexpression of TMSB4XP8 has no effect on CCSER1. We hypothesize that it is a local process, in cis, possibly through a polymerase clashing. We have added a paragraph in the discussion session, and additional references, to explain in more detail this proposed model.

In Fig. 3, the authors introduced the rearranged forms of CCSER1 (rCCSER1) into cells and tested the cell proliferation and other phenotype conferred by rCCSER1. Please provide the exact structure of rCCSER1. Which are the missing exons?

We isolated the isoform from the SW-620 cell line (below an IGV representation). We would be happy to include also the corresponding sequence in the Supplementary data.

[Figure removed by editorial staff per authors' request]

The authors presented the data of Aurora Kinase inhibitor in Fig. 3h. Unlike Fig. 3a, the experiments were terminated at 72 hours. Since the difference was small and KO TMS cells with the inhibitor seemed to accelerate the growth after 48 hours, this reviewer requests the experiment with an extended period of time, up to 96 hours, the same as Fig. 3a.

Experiments presented in figure 3a and 3h have been performed on different cell lines (Hela and RPE-fig3a and DLD1-fig 3h), so the growth kinetic of the cells is different. We were not able to find a condition that prevented reaching confluence at 96h while maintaining a reliable growth curve (cells suffered higher dilutions and did not grow evenly when plated at low number). A representative experiment is shown below.

Other comments

Abstract

"FRA4F, which is present in up to 18% of cancer cases" - either "which is expressed" or "which is deleted." FRA4F is present in the human genome.

We corrected it.

Results and Discussion

Page 3, "only one gene significantly deregulated" - "significantly deregulated" here seems redundant, as you mentioned it in the next sentence.

We removed the "significantly" from the following sentence.

Figure 1 - in somewhere, please provide the genomic view (UCSC genome browser view) of *FRA4F*, so that we can understand how large the CFS (and *CCSER1* gene) is.

A Supplementary figure has been added as requested (Suppl. Figure 2)

Figure 1d - please spell out all the tumor types somewhere in the legend.

As the list is long, we added a supplementary table with the labels explained.

Page 5, "this genetic lesion represents the most pervasive" - "lesion" should be "region"

We have modified the sentence to make it more clear.

Page 5, "Given the widespread involvement of *FRA4F* in cancer" - unclear. "Given the frequent deletion of *FRA4F* in cancer"

We have modified the sentence accordingly.

Page 6, "To this end, we designed sgRNA probes" - "probes" is not necessary.

Corrected.

Page 9, "preserve the expression of in-frame *CCSER1* mRNA" - unclear. "express the *CCSER1* in-frame variants"

Corrected.

Figure 3a, can you make the Fig. 3a bigger? The sizes of the font in 3a are too small, compared to other figures in Fig. 3.

Corrected.

Figure 3c, gammatub needs attention.

We kindly ask the Reviewer to provide more details on this point.

Page 12, "*CCSER1* deficiency has been associated with cell division defects" - Cell cycle profiling by FACS would be powerful to strengthen your argument.

While we do not detect major differences in the cell cycle profiling, we are now enclosing a movie showing disrupted mitosis upon r-*CCSER1* overexpression (Supplementary movie).

Page 12, Please explain briefly what are the *CIN25* and *CIN70* better. Also, only *CIN25* is presented in Fig. 3G.

We have added an explanation of *CIN25* and *CIN70* in the main text. Also, here is the representation of *CIN70*, we would be happy to include it in the main figures, or as supplementary figure.

May 24, 2021

RE: Life Science Alliance Manuscript #LSA-2021-01019-TR

Dr. Giovanni Tonon
San Raffaele Scientific Institute
Functional Genomics of Cancer Unit, Division of Molecular Oncology
Via Olgettina 60
Milan 20032
Italy

Dear Dr. Tonon,

Thank you for submitting your revised manuscript entitled "Deletion of a pseudogene in a fragile site triggers oncogenic expression of the CCSE1 mitotic gene". We would be happy to publish your paper in Life Science Alliance pending final revisions necessary to address the minor concerns raised by Rev 1 and to meet our formatting guidelines.

Please also attend to the following,

- please make sure that titles in the system and in the manuscript text match
- please consult our manuscript preparation guidelines <https://www.life-science-alliance.org/manuscript-prep> and make sure your manuscript sections are in the correct order
- please separate the Results and Discussion section into two - 1. Results 2. Discussion, as per our formatting requirements
- please use Capital letters when introducing the panels in the figures, callouts in the manuscript text, and in figure legends (e.g. instead of a please use A, etc.)
- please add your main, supplementary figure, and table legends to the main manuscript text after the references section
- please upload your main and supplementary figures as single files
- please be sure to add all Authors in the Authors' Contribution section in the manuscript text
- please add scale bars to Figures 2C, 3C-F

A. FINAL FILES:

B. MANUSCRIPT ORGANIZATION AND FORMATTING:

Sincerely,

Shachi Bhatt, Ph.D.
Executive Editor

Reviewer #1 (Comments to the Authors (Required)):

The authors have addressed most of the issues and/or specified unclear points raised in my previous review.

I appreciate that they have addressed some of these issues in the discussion and specified unclear points in their rebuttal letter, however, some of these clarifications should be clearly stated in the text and added to the manuscript to convey the readers a more precise and complete message:

1) For example, the authors state in their rebuttal letter that "Out of the 6142 analyzed events from TCGA involving CCSER1, 3155 (51.37%) are also overlapping TMSB4XP8. It should be mentioned that we found that deletion of CCSER1, irrespectively on TMSB4XP8 engagement, consistently lead to the increase in CCSER1 expression, suggesting that other mechanisms are involved, either indirectly impacting on TMSB4XP8, or through alternative paths".

This should also be discussed in the text, together with the fact that within the boundaries of the deletions not encompassing TMSB4XP8 other elements, like pseudogenes and ncRNAs, may alter the expression of CCSER1. In addition, the deletions may also affect splicing (Bryen SJ et al., Am J Hum Genet. 2019).

2) Please add the data showing the effect of the short, long or mutant isoform overexpression on HeLa and RPE-1 cell proliferation and indicate whether the effect is statistically significant.

3) It would be useful to provide the exact structure of the mutated isoform showing which exons are included or missing.

4) Please specify the cell line in which the quantification of multinucleated cells after expression of short, long, or rearranged isoform has been done and add this data to the manuscript.

5) The authors state "we would argue that the overexpression of CCSER1 transcripts, irrespectively from the size of the in-frame deletion, would always lead to increased proliferation and sensitivity to Aurora kinase inhibitors, as even the WT isoforms trigger mitotic instability upon overexpression". To support this conclusion, the authors should treat the cells expressing the WT or mutant isoforms with the Aurora A kinase inhibitor(s), particularly in light of the fact that different inhibitors may have a different impact depending on which isoform is expressed.

6) I thank the authors for explanation of the graph showing the % of extracentrosomes. Since the authors observe an increase in the centrin signal in the r-CCSER1 overexpressing cells, not present in cells transfected with the mock, where the centrin and gamma-tubulin always colocalized, the authors should keep this graph and explain in the text or the legend how they quantified extracentrosomes, ex. mitotic figures showing ectopic/not colocalized centrin/gamma tub signals.

7) Please specify also the sizes of the bands corresponding to the non-deleted regions in Suppl Fig. 3.

Minor points:

There are some typos, ex. FR6H (page 3, line 46 and page 4 Fig.1 legend), TMBS4XP8 (Page 8 Fig.2 legend).

Reviewer #2 (Comments to the Authors (Required)):

The authors addressed my concerns (mostly).

Milan, June 12th, 2021

Dear Dr. Bhatt,
many thanks for your help, as well as to Reviewer 1, for her/his constructive comments, as we believe that now the manuscript has much improved.
Please find enclosed our responses to the Reviewer points.

Many thanks again.
Kind regards,
Giovanni Tonon

Reviewer #1 (Comments to the Authors (Required)):

The authors have addressed most of the issues and/or specified unclear points raised in my previous review.

I appreciate that they have addressed some of these issues in the discussion and specified unclear points in their rebuttal letter, however, some of these clarifications should be clearly stated in the text and added to the manuscript to convey the readers a more precise and complete message:

1) For example, the authors state in their rebuttal letter that "Out of the 6142 analyzed events from TCGA involving CCSER1, 3155 (51.37%) are also overlapping TMSB4XP8. It should be mentioned that we found that deletion of CCSER1, irrespectively on TMSB4XP8 engagement, consistently lead to the increase in CCSER1 expression, suggesting that other mechanisms are involved, either indirectly impacting on TMSB4XP8, or through alternative paths". This should also be discussed in the text, together with the fact that within the boundaries of the deletions not encompassing TMSB4XP8 other elements, like pseudogenes and ncRNAs, may alter the expression of CCSER1. In addition, the deletions may also affect splicing (Bryen SJ et al., Am J Hum Genet. 2019).

We agree with the Reviewer and have added in the text (page 9, line 223, in red) a discussion concerning this point, including the suggested reference as well.

2) Please add the data showing the effect of the short, long or mutant isoform overexpression on HeLa and RPE-1 cell proliferation and indicate whether the effect is statistically significant.

We have now added the data and stat relative to proliferation increase upon overexpression of short, long or rearranged isoform on HeLa and RPE cell lines (see figure 3), and modified the figure legend accordingly.

3) It would be useful to provide the exact structure of the mutated isoform showing which exons are included or missing.

The Reviewer is right, we have added a supplemental figure 4 reporting the sequence and included a sentence in the Materials and Methods (page 11, line 274, in red): The construct includes exons 2 (where the ATG is located) to exons 5

included, and then exon 11 (Supplemental Figure 4). Notably, the splicing between exon 5 and 11 creates an isoform which is not in frame, thus only a portion, out of frame, of exon 11 is translated.

4) Please specify the cell line in which the quantification of multinucleated cells after expression of short, long, or rearranged isoform has been done and add this data to the manuscript.

We have added the data as requested, as well as indicated in the figure legend the cell line used (HeLa sgRNA for CCSER1).

5) The authors state "we would argue that the overexpression of CCSER1 transcripts, irrespectively from the size of the in-frame deletion, would always lead to increased proliferation and sensitivity to Aurora kinase inhibitors, as even the WT isoforms trigger mitotic instability upon overexpression". To support this conclusion, the authors should treat the cells expressing the WT or mutant isoforms with the Aurora A kinase inhibitor(s), particularly in light of the fact that different inhibitors may have a different impact depending on which isoform is expressed.

We thank the Reviewer for this suggestion. We decided not to pursue these experiments early on during the project, as we considered the experiments conducted in a cell line, like the DLD1, where CCSER1 is already overexpressed and so adjusted to its overexpression, more informative to the potential clinical use of aurora kinase inhibitors in the clinic than using cell lines overexpressing CCSER1.

6) I thank the authors for explanation of the graph showing the % of extracentrosomes. Since the authors observe an increase in the centrin signal in the r-CCSER1 overexpressing cells, not present in cells transfected with the mock, where the centrin and gamma-tubulin always colocalized, the authors should keep this graph and explain in the text or the legend how they quantified extracentrosomes, ex. mitotic figures showing ectopic/not colocalized centrin/gamma tub signals.

We thank the Reviewer for this important observation. We have now added the suggested explanation in the figure legend.

7) Please specify also the sizes of the bands corresponding to the non-deleted regions in Suppl Fig. 3.

We have added additional labels in the figure and the exact predicted size in the figure legend.

Minor points:

There are some typos, ex. FR6H (page 3, line 46 and page 4 Fig.1 legend), TMBS4XP8 (Page 8 Fig.2 legend)

We thank the Reviewer, we have corrected the typos (in red).

June 14, 2021

RE: Life Science Alliance Manuscript #LSA-2021-01019-TRR

Dr. Giovanni Tonon
San Raffaele Scientific Institute
Functional Genomics of Cancer Unit, Division of Molecular Oncology
Via Olgettina 60
Milan 20032
Italy

Dear Dr. Tonon,

Thank you for submitting your Research Article entitled "Deletion of a pseudogene in a fragile site triggers oncogenic expression of the CCSE1 mitotic gene". It is a pleasure to let you know that your manuscript is now accepted for publication in Life Science Alliance. Congratulations on this interesting work.

DISTRIBUTION OF MATERIALS:

Again, congratulations on a very nice paper. I hope you found the review process to be constructive and are pleased with how the manuscript was handled editorially. We look forward to future exciting submissions from your lab.

Sincerely,
